# Malnutrition with Low Muscle Mass Is Common after Weaning off Home Parenteral Nutrition for Chronic Intestinal Failure

**DOI:** 10.3390/nu15020338

**Published:** 2023-01-10

**Authors:** Lucas Wauters, Solène Dermine, Brune de Dreuille, Joanna Bettolo, Coralie Hutinet, Ashiq Mohamed, Emilie Lecoq, Lore Billiauws, Alexandre Nuzzo, Olivier Corcos, Francisca Joly

**Affiliations:** 1Department of Gastroenterology and Hepatology, University Hospitals Leuven, 3000 Leuven, Belgium; 2Department of Gastroenterology and Nutrition Support, APHP Beaujon Hospital, 92110 Clichy, France; 3Institut National de la Santé et de la Recherche Médicale (INSERM), University of Paris, UMR 1149, 75890 Paris, France; 4Laboratoire de Recherche Vasculaire Translationnelle (LVTS), Institut National de la Santé et de la Recherche Médicale (INSERM), Université Paris Cité, Bichat Hospital, U1148, 75018 Paris, France

**Keywords:** intestinal failure, parenteral nutrition, short bowel syndrome, malnutrition, sarcopenia, teduglutide

## Abstract

The differences in outcomes after weaning off intravenous support (IVS) for chronic intestinal failure (IF) are unclear. Adult IF patients who are weaned off IVS at a tertiary care center (June 2019–2022) were included in this study, and nutritional and functional markers were assessed before, during, and after weaning. Short bowel syndrome (SBS) was present in 77/98 of the IF patients, with different outcomes according to the final anatomy. The body weight and the BMI increased during IVS in those with a jejunocolonic (JC) anastomosis (*p* < 0.001), but weight loss was significant during follow-up (*p* < 0.001). Malnutrition was present in >60%, with a reduced muscle mass, which was found using bioelectrical impedance analysis (BIA), in >50% of SBS-JC patients. Although reduced hand-grip strength and sarcopenia were less common, the muscle quality, or phase angle (BIA), decreased during follow-up, also correlating with serum albumin and muscle mass (*p* ≤ 0.01). The muscle quality and albumin were low in the patients restarting IVS, which was only the case with ≤60 cm of small bowel. Closer follow-up and earlier treatment with teduglutide (TED) should be considered in these patients, as none of the TED-treated patients were malnourished or sarcopenic. Studies on the potential benefits of nutritional and physical interventions for low muscle mass and associations with outcomes are needed in chronic IF patients.

## 1. Introduction

Chronic intestinal failure (IF) is defined as “the persistent reduction of gut function below the minimum necessary for the absorption of macronutrients and/or water and electrolytes, such that intravenous supplementation (IVS) is required to maintain health and/or growth” [1]. Patients are provided IVS through home parenteral nutrition (HPN) programs [2]. Despite its chronic nature, the need for HPN may be reversible, with non-transplant or reconstructive surgery as a common cause of weaning off HPN in a recent one-year international multicenter survey [3]. This is especially the case for short bowel syndrome (SBS), which is the most common cause of chronic IF [4]. SBS is defined as <2 m of remaining small bowel (SB) after surgical resection; however, it can also occur due to an impaired function of >2 m of the SB [5]. The restoration of intestinal continuity (RC) is indicated, if possible, as this will increase the odds of weaning off HPN and regaining full nutritional autonomy [6,7,8,9].

After reconstructive surgery, the probability of weaning off HPN is higher in the presence of the colon and ileocecal valve in SBS [6,10]. Besides the increased absorption, bringing a colon into continuity induced structural and functional adaptive changes [11]. Although the process of intestinal adaptation may result in nutritional autonomy, or weaning off HPN, the body mass index (BMI) decreased during HPN and was only slightly higher in HPN-independent SBS patients [6]. In addition, no differences were found in the muscle mass between the SBS patients with or without a remnant colon [12,13]. As BMI underestimates the presence and the degree of low muscle mass, it is important to measure the body composition in chronic IF patients [13]. Besides malnutrition, sarcopenia, or a low muscle mass and strength, was frequently found in IF patients using different methods [13,14]. While sarcopenia was also present in those with only oral nutritional supplements or intestinal insufficiency, its evolution during and after weaning off IVS has not been assessed.

Indeed, studies on the nutritional status of chronic IF patients after weaning are scarce. Although a structured and multidisciplinary approach has been advocated in order to increase nutritional autonomy and reduce morbidity and mortality in specialist centers, there are only few longitudinal studies regarding weaning off HPN. Therefore, we aimed to study the evolution in nutritional and functional markers both during and after weaning off HPN in chronic IF patients. We also assessed the prevalence of malnutrition and sarcopenia, using both existing and novel markers of body composition. Special attention was paid to the final anatomy in SBS patients, as nutritional autonomy may be achieved immediately or within two years after reconstructive surgery in those with a preserved colon [6,15]. Finally, the effects of novel treatments with glucagon-like peptide (GLP-)2 analogues were evaluated.

## 2. Materials and Methods

### 2.1. Study Population

All consecutive adult patients who were weaned off HPN at a tertiary referral center (CHU Beaujon, Clichy, France) between June 2019 and June 2022 were identified from a prospectively maintained database (IRB approval number CER-2021-73). Patients with an IVS duration of >1 month and without chronic infections (e.g., HIV) or primary malignant causes of HPN were included, which is similar to previous studies [6,15]. The development of new malignancies and the use of systemic anti-cancer therapy or death during HPN were recorded. The reasons for weaning off IVS were restoration of continuity (RC) or spontaneous adaptation with standard dietetic and medical treatment [1]. Since the marketing authorization of teduglutide (TED) in France (October 2015), enhanced adaptation was also possible in SBS patients with ongoing IVS, despite standard treatment [1]. All TED-treated patients received a dose of 0.05 mg/kg/day subcutaneously.

HPN was delivered according to international guidelines [16]. The composition, frequency, and volume of IVS were adjusted to individual needs during regular assessments (see below). As previously described, the minimal level of IVS was determined by monitoring the nutritional status, including hydration (urine output), body weight (BW) and serum albumin levels, creatinine, and electrolytes. In cases of nutritional stability, a reduction in the frequency and/or volume of IVS was gradually performed until weaning. In contrast, weight loss and decreased serum albumin levels led to an increase to previous levels or even the restarting of IVS in patients who were already weaned [6,10,17].

### 2.2. Definitions

The SBS subgroups were based on final anatomy. Temporary SBS was defined as >2 m of SB with a colon-in-continuity after reconstructive surgery or RC. Patients with <2 m of SB after resection were classified as jejuno-colic (SBS-JC) or jejuno-ileal anastomosis (SBS-JIC) with colon-in-continuity [4]. Patients with >2 m of SB, but with an impaired function and no remnant colon, were classified as ileostomy (SBS-I) [5]. Remnant SB length was measured from the ligament of Treitz during surgery and colon-in-continuity was expressed as a percentage, according to Cummings et al. [18]. Other pathophysiological mechanisms of IF were intestinal dysmotility, e.g., chronic intestinal pseudo-obstruction (CIPO), mechanical obstruction, fistulas, and extensive SB mucosal diseases. Finally, the severity of IF was defined according to the ESPEN categories, based on the daily mean of the maximum IVS volume infused per week [4].

### 2.3. Data Collection

The patient demographics and clinical characteristics were collected from medical records, including the dates of IVS initiation and withdrawal. This also included any previous need for IVS and/or enteral nutrition (EN) or a need to restart IVS during follow-up (FU). Patients were routinely evaluated every 3 to 6 months during and after weaning off IVS at the outpatient clinic, and data from both the first (start) and the final date of IVS (stop), and the longest available duration of FU, were used. Besides BW and BMI, blood tests included serum albumin levels, creatinine, electrolytes, and C-reactive protein (CRP). Micronutrients (trace elements and vitamins) were measured every 6 months to 1 year and during FU after weaning. As of June 2021, body composition was determined using bioelectrical impedance analysis (BIA) according to the manufacturer’s instructions (Bodystat 1500, Bodystat, Douglas, Isle of Man). To account for differences in height, fat-free mass index (FFMI) was calculated by dividing fat-free mass by height squared, and muscle quality was determined using phase angle (PA). PA is an indicator of cell membrane health and integrity, with higher values reflecting better cell function and muscle mass. Hand-grip strength (HGS) was measured using a dynamometer (Jamar, FysioSupplies, Antwerp, Belgium) by trained dietitians, and the highest value of both hands was recorded following the recommendations [19].

### 2.4. Statistical Analysis

Continuous data were reported as median (interquartile range, or IQR) and categorical data as proportions, with non-parametric testing to account for not-normally distributed variables. The baseline differences between the groups were compared with Kruskal–Wallis tests for continuous data and chi-square tests for proportions. In addition, Dunn’s multiple comparison test was used when comparing continuous data between individual groups, with correction for multiple testing. Changes in nutritional markers during IVS and FU after weaning were analyzed using 1-way analysis of variance (between-groups) and Wilcoxon tests (within-group). Malnutrition after weaning was based on weight loss, low BMI, and/or FFMI using the Global Leadership Initiative on Malnutrition (GLIM)-criteria, with the same cut-offs for FFMI (<15 kg/m^2^ for women and <17 kg/m^2^ for men) as the earlier diagnostic criteria of malnutrition [20,21]. Reduced HGS (<16 kg for women and <27 kg for men) and sarcopenia were based on the revised European consensus [22], with additional use of age- and gender-specific cut-offs for HGS [19]. Finally, correlations between the variables of interest were determined using the Spearman correlation coefficient. For all analyses, two-tailed *p*-value < 0.05 were considered significant, and 0.05 < *p* < 0.1 were considered a trend. Data were analyzed in Prism GraphPad, version 8.0 (Graphpad Software, San Diego, CA, USA).

## 3. Results

### 3.1. Study Population

In total, 98 IF patients were included, with their demographics and clinical characteristics shown in Table 1. The median IVS duration was 31 weeks, with a median follow-up of 41 weeks after weaning. The main mechanism of IF was SBS, followed by dysmotility, mechanical obstruction, mucosal disease, and fistulas (Table 1). The causes of SBS were arterial or venous mesenteric infarction, inflammatory bowel disease (IBD), surgical complications, radiation enteropathy, and trauma or volvulus (see below). The causes of dysmotility were CIPO (*n* = 6), narcotic bowel (*n* = 2), and small bowel diverticulosis (*n* = 1). Mechanical obstruction was caused by small bowel stenosis after gastrectomy (*n* = 3), bypass surgery, and radiation enteropathy (both *n* = 1). Mucosal diseases (*n* = 4) and fistulas (*n* = 3) were caused by enteritis (mainly Crohn’s disease); however, these were excluded because of their low numbers. Due to the development of progressive malignancies or death (unrelated to HPN), six SBS patients were also excluded from the analysis (two temporary SBS, two SBS-JC, and two SBS-JIC patients) (Figure 1). The SBS patients who were weaned on TED (four SBS-JC patients and one SBS-I patient) were analyzed separately. After weaning, one SBS-JIC patient and one dysmotility patient (narcotic bowel) were lost to FU. Thus, a total of 70 SBS and 83 IF (including dysmotility and obstruction) patients completed FU and were included in the analyses.

### 3.2. SBS Patients

#### 3.2.1. Patient Characteristics

The demographics and the clinical characteristics of the SBS subgroups are shown in Table 2. While the age was similar, a lower percentage of females was found in SBS-I and TED-treated patients. The presence of IBD and post-surgical complications was more common in SBS-I and TED-treated patients, respectively. The use of fluid and electrolytes (FE for SBS-I) and the highest daily mean volume of energy (PN4 for TED) was also more common, but the number of patients was low. Except for SBS-I, the total SB length was higher in the temporary vs. the definite SBS patients (all p_adj_ < 0.0001), which is consistent with the definition and shortest IVS duration in temporary SBS (see below). The ileal length and the colon-in-continuity were similar in the SBS-JIC vs. the temporary SBS patients. In contrast, the ileal length and the colon-in-continuity were lower in the SBS-JC vs. the SBS-JIC and the temporary SBS patients (all p_adj_ ≤ 0.001). The patients on TED had similar ileal length and colon-in-continuity vs. the SBS-JC patients.

#### 3.2.2. Evolution during IVS

The daily mean volumes of energy (PN1-PN4) were similar between the SBS patients without TED, which is similar to that previously described [4]. Based on the presence of >2 m SB with a colon-in-continuity, weaning was possible after RC in temporary SBS patients with an IVS duration of <12 weeks (Table 2). While three patients were weaned before RC (Figure 1), those with postoperative complications due to fistulas (*n* = 3), infections (*n* = 2), or additional resections (*n* = 1) had a longer IVS duration (37 (30.5; 40) weeks, *p* < 0.0001). Compared to temporary SBS patients, IVS duration was longer in definite SBS patients (all p_adj_ < 0.05). This was clearer when comparing IVS duration following RC, which was longer in SBS-JC (26 (13; 45) weeks, p_adj_ < 0.0001), SBS-JIC (13 (3; 45) weeks, p_adj_ < 0.01), and TED-treated patients (67 (8; 90) weeks, p_adj_ < 0.0001) compared to temporary SBS patients (median 1 (0; 3) weeks after RC).

Both BW (Δ = 4.1 (0.6; 10) kg) and BMI (Δ = 1.5 (0.2; 3.7) kg/m^2^, both *p* < 0.0001) increased during IVS in all of the SBS patients (*n* = 70). However, the evolution differed between the groups (Table 3), with a significant increase in the SBS-JC patients (BW Δ = 5 (1.5; 9.8) kg and BMI Δ = 1.8 (0.5; 3.1) kg/m^2^, both *p* < 0.001) and trends for SBS-JIC, SBS-I, and TED-treated patients (all *p* < 0.1). In addition, the serum albumin increased (Δ = 11.6 (4.6; 17.4) g/L) and the CRP decreased (Δ = −1 (0; −10) mg/L, both *p* < 0.0001) during IVS in all of the SBS patients, with no between-group differences (Table 3). The results were similar when excluding those with previous or intermittent HPN (four SBS-JC patients, one SBS-JIC patient, and one SBS-I patient), FE (two temporary SBS and three SBS-I patients), and/or EN (five temporary SBS patients) (*n* = 55).

#### 3.2.3. Evolution during FU

Compared to temporary SBS, the duration of FU was longest in the TED-treated patients (p_adj_ < 0.05). The IVS duration before TED was 133 (77.3; 267) weeks, with 33.3 (6.9; 47.3) weeks of treatment until weaning. No change in BW or BMI was seen during FU in any of the SBS patients (*n* = 70). However, the evolution differed between the groups (Table 3), with a significant increase in the temporary SBS patients (BW Δ = 3.5 (−1.3; 10.5) kg and BMI Δ = 1.3 (−0.5; 3.6) kg/m^2^, both *p* < 0.01), but a decrease in the SBS-JC patients (BW Δ = −6 (−13; −2.5) kg and BMI Δ = −2.1 (−4.8; −0.9) kg/m^2^, both *p* < 0.001). While albumin (Δ = 2.3 (−1.2; 5.7) g/L, *p* = 0.003) increased during FU in all of the SBS patients, this was only found in temporary SBS (albumin Δ = 3.2 (0.9; 6.7) g/L, *p* = 0.003). In contrast, decreased BW (Δ = −3 (−7; 3) kg) and BMI (Δ = −0.9 (−2.5; 1) kg/m^2^, both *p* = 0.04) were found when excluding those with previous or intermittent HPN, FE, and/or EN, but with a similar increase in albumin (*n* = 55).

During FU, IVS was restarted after 13 (9; 15) weeks in three SBS-JC patients and after 35 weeks in one SBS-JIC patient (Figure 1). The duration of IVS was 39 (35; 54) weeks and 79 weeks, respectively. When these patients were excluded, a similar decrease in BW and BMI was found in the SBS-JC patients during FU, but with increased albumin levels (Δ = 3.2 (−0.3; 4.1) g/L, *p* = 0.02).

Similarly, albumin increased in the SBS-JIC patients with no need to restart IVS (Δ = 5.5 (1.5; 9) g/L, *p* = 0.02). Despite stable CRP levels during FU, the albumin levels decreased (<30 g/L) in those restarting IVS (Figure 2), with important deficiencies in micronutrients (Table 4). Of note, all of the SBS patients restarting IVS had ≤60 cm of total SB (40 (15; 60) cm of jejunum in SBS-JC and 30 cm of jejunum, plus 10 cm ileum, in SBS-JIC). All of the stable SBS patients during FU after weaning had a jejunal and total SB length above this cut-off point.

### 3.3. Other Causes Than SBS

For dysmotility (*n* = 8), weaning was possible in one patient after surgery (diverticulosis), with intermittent HPN in two patients (narcotic bowel). Of the six CIPO patients with complete FU (50% female, 55 (50–65) years old), one restarted IVS during FU because of malnutrition (see below). During IVS, both BW (Δ = 6 (3; 13.4) kg) and BMI (Δ = 2.4 (1.1; 4.4) kg/m^2^, both *p* < 0.01), as well as albumin (Δ = 8.5 (3.3; 13.3) g/L, *p* = 0.02), increased. No changes were found during FU. In the patients with mechanical obstruction (*n* = 5), weaning was only possible after reconstructive surgery or endoscopy (e.g., dilatation of anastomotic stenosis), with temporary EN in two patients. However, no changes in BW, BMI, or albumin were found during IVS or FU.

### 3.4. Malnutrition and Sarcopenia

Following GLIM-criteria, malnutrition was present in 33 (47.1%) SBS patients (*n* = 70) during FU, with no differences between the subgroups. Similar proportions of SBS-JC (13/21 or 61.9%) and SBS-JIC patients (7/12 or 58.3%) were malnourished, of which six (54.5%) SBS-JC and four (66.7%) SBS-JIC patients with available BIA had a low FFMI (Figure 3). However, malnutrition was solely based on a low FFMI in one (7.7%) SBS-JC patient and two (16.7%) SBS-JIC patients. Moreover, only one (5.6%) SBS-JC patient and none of the SBS-JIC patients had a reduced muscle strength or sarcopenia, according to the revised European consensus [22]. When considering age- and gender-specific cut-offs for HGS, sarcopenia was present in three (16.7%) SBS-JC and three (27.3%) SBS-JIC patients [19]. Of note, none of the SBS-JC patients on TED were malnourished or sarcopenic.

In contrast, low muscle quality, or PA, was present in ten (55.6%) SBS-JC and three (27.3%) SBS-JIC patients. Interestingly, PA also decreased during FU (Δ = −0.8 (−1.9; −0.4), *p* = 0.03) and correlated with albumin (r = 0.6, *p* < 0.01) and FFMI (r = 0.58, *p* = 0.01) in the SBS-JC patients. Moreover, PA was low in patients restarting IVS. While FFMI and HGS were normal, HGS increased only in those with no need to restart IVS during FU (Δ = 6.7 (0.9; 8.7) kg for SBS-JC and Δ = 5.7 (3.9; 11) kg for SBS-JIC, both *p* < 0.01). Increased HGS was also found in all of the IF patients (Δ = 4.4 (−0.7; 8.4) kg, *p* < 0.0001), including temporary SBS (Δ = 4.4 (−1; 8.4) kg) with increased FFMI (Δ = 0.6 (0.15; 1.5), both *p* < 0.01). The temporary SBS patients with malnutrition (8/26 or 30.8%) had a low FFMI, with a low HGS or sarcopenia in two (8.7%) patients, or six (26.1%) when using age- and gender-specific cut-offs [19]. Overall, 40 (48.2%) of the 83 IF patients were malnourished, with a similar proportion for dysmotility (50%), of which all had low FFMI, and sarcopenia was found in one CIPO patient who restarted IVS.

## 4. Discussion

### 4.1. Summary

Chronic IF represents an important clinical challenge, due to the complexity of underlying diseases and the need for multidisciplinary management. Despite the potential for weaning off HPN, the evolution in both nutritional and functional markers has not been systematically assessed. In this analysis of adult IF patients who followed and were weaned off HPN in a tertiary care center, we showed that the outcomes differed according to the cause and the final anatomy for SBS. The patients with >2 m of SB and colon-in-continuity after reconstructive surgery showed beneficial outcomes, including increased BW, BMI, albumin, FFMI, and HGS after weaning. In comparison, the SBS-JC patients were at risk of weight loss after weaning, with a >60% prevalence of malnutrition, including a low FFMI. However, the prevalence of reduced HGS and sarcopenia was limited. HGS even increased during FU after weaning, including in the SBS-JC and SBS-JIC patients with no need to restart IVS. On the contrary, the muscle quality, or PA, decreased and correlated with albumin, which was also reduced in those restarting IVS. Therefore, the measurement of PA may prove useful in the FU of these patients and the use of GLP-2-analogues should especially be considered in the cases of a total SB length of ≤60 cm, as none of the TED-treated patients were malnourished or sarcopenic.

### 4.2. SBS Anatomy and Evolution during IVS and FU

In a recent one-year international survey, weaning off HPN was associated with non-transplant or reconstructive surgery in 41.9% of patients, spontaneous adaptation in 50.7% of patients, intestinal transplantation in 5.1% of patients, and intestinal growth factor therapy in 2.2% of patients [3]. Indeed, the restoration of continuity, or RC, is indicated, if possible, as this may lead to only temporary SBS. In this analysis, SBS subgroups were based on the final anatomy, which is required in order to correctly compare the outcomes [23]. Similar to previous studies, the proportion of definite end-ostomy patients was small compared to SBS-JC and SBS-JIC patients [6,10]. This was also reflected by the higher proportions of mesenteric ischemia, for which RC is performed if it is feasible after a period of 3–6 months, according to the ESPEN-guidelines [24]. Using the current definition of temporary SBS, it was possible to distinguish those who were weaned after (or even before) reconstructive surgery and with the use of EN in some. While an end-ostomy as a temporary solution led to beneficial outcomes [5,15], a longer IVS duration was needed in those with post-operative complications (e.g., fistulas). Nevertheless, increased BW and albumin were found in all of the temporary SBS patients during FU, with even increased FFMI or muscle mass in those with repeated BIA. Compared to the previous findings of CT-defined muscle gain during HPN [13], these are the first prospective data on the body composition of SBS patients after weaning from HPN.

Despite the increased BW and BMI during IVS, the SBS-JC patients were at risk of weight loss after weaning. Due to the lower ileal length and colon-in-continuity compared to SBS-JIC and temporary SBS patients, both IVS dependency, as well as the evolution after weaning, may depend on the presence of the ileum and/or colon. Besides fluid and electrolyte absorption, the colonic fermentation of carbohydrates and the absorption of short-chain fatty acids and medium-chain triglycerides provide additional calories [25,26]. As changes in IVS calories correlated with both BMI and fat-free mass (FFM), which was determined using dual X-ray absorptiometry (DXA), it is likely that insufficient colonic energy absorption led to both weight loss and reduced muscle mass after weaning [27]. Indeed, low FFMI was found in >50% of malnourished SBS-JC patients. The reduced FFM is in agreement with previous studies, showing no protective effect from an incomplete colon in SBS [12,13]. Although the prevalence of reduced muscle strength and sarcopenia was low, this increased when using age- and gender-specific cut-offs. Nevertheless, HGS increased during FU in all of the patients, including the SBS-JC and the SBS-JIC patients, with no need to restart IVS. Conversely, the muscle quality, or PA, decreased in those restarting IVS. Of note, the use of PA is more reliable than other markers and predictive of the outcome in IF and other patient populations [28,29]. Moreover, PA correlated with albumin, which decreased in the patients restarting IVS, despite stable CRP, which is important for the interpretation of nutritional deficiencies. The link with low albumin levels and nutritional risk in hospitalized subjects also suggest its value to monitor changes during FU and after nutritional interventions [30].

### 4.3. SB Length, Treatment or IF Causes, and Need for FU

Besides decreased albumin, important mineral and/or vitamin deficiencies after weaning were the reasons for restarting IVS in patients with ≤60 cm of SB. Although some level of intestinal adaptation is possible during the first two years after diagnosis, the limited jejunal length is in agreement with the previously proposed cut-off of 65 cm [10]. The lack of ileum and colon leads to lower levels of pro-adaptive hormones, of which GLP-2 is the best studied [31,32]. Treatment with TED is beneficial in these patients, with 12% weaning off HPN in phase-3 and open-label extension studies, of which 75% had a colon-in-continuity [33]. Despite a median SB length of ≤60 cm (52.5 cm), a stable nutritional status was noted after weaning [33]. Besides the lower response but higher weaning rates in those with a remnant colon, the highest IVS volume increase within one year from discontinuation of TED was found in the SBS-JC patients [34,35]. Although the response rates of TED are based on improved wet weight absorption and IVS volume reduction, the outcomes are also dependent on colonic energy absorption. Interestingly, none of the SBS-JC patients on TED were malnourished or sarcopenic during FU, pointing to increased energy and protein absorption. While studies with markers of energy absorption (e.g., metabolic balance studies) are needed in relation to body composition, a longer period of HPN may have been needed in those with ≤60 cm of SB to adapt [15]. Interestingly, the use of TED as part of postsurgical treatment predicted nutritional autonomy, with beneficial outcomes for ultra-short bowel syndrome (SB length of ≤50 cm) [8,36]. In contrast, FU every three to six months seems to be needed in those with malnutrition based on GLIM-criteria, including low muscle mass, especially in SBS-JC and/or patients with ≤60 cm of SB.

Considering other causes, the dysmotility patients showed improved nutritional status during HPN, similarly to that previously observed [17]. Although reversibility of IF and weaning was less common for CIPO in a multicenter and international study [15,17], the prevalence of malnutrition was similar in this study, and only one CIPO patient with sarcopenia restarted HPN. No specific recommendations regarding FU can be made based on the lower number of CIPO patients. Increased body weight and albumin were found in post-bariatric surgery patients with IF, which can be caused by dysmotility and mechanical obstruction [37]. However, weaning was only possible after reconstructive surgery, which is similar to this study with temporary EN in two patients. Of note, the majority of issues observed in HPN patients post-bariatric surgery were due to late complications or malnutrition in an ESPEN survey, including low albumin and micronutrient deficiencies. As the numbers of patients with mucosal diseases is often too limited for analysis, this was previously grouped as miscellaneous diseases with lower odds of weaning in a recent study [3].

### 4.4. Strenghts and Limitations

The strengths of this study include the comprehensive nutritional assessment of adult IF patients before, during, and after weaning off HPN. Although BIA was not available for all of the patients or visits, we present the first longitudinal data for changes in the body composition of adult IF patients after weaning. Indeed, an altered body composition with low FFM and micronutrient abnormalities have also been reported in children after weaning [38]. Although the treatment and the FU at a tertiary care center may limit the generalizability of our findings, the care for IF is highly specialized and our population is similar to that of previous surveys [4]. While we included different causes of IF, the number of patients was lower in some subgroups. We did not include patients with ongoing HPN, as differences in the outcomes, including HPN dependency, are already available from our center and are known for the SBS subtypes, including lower odds of weaning for SBS-JC patients [3,6,10]. The results were robust when excluding patients with previous or intermittent HPN, FE only, and EN, with a separate analysis in those restarting IVS. Nevertheless, more data are needed on the use of BIA equations and cut-offs in IF patients. Finally, the use of grip strength as a functional assessment may be of limited value in this (younger) population and only a supportive and not a core phenotypic measure of the GLIM-criteria for malnutrition [21].

## 5. Conclusions

The evolution of nutritional outcomes differed during and after weaning off HPN in adult IF patients who were followed at a tertiary care center, with a particular risk of weight loss in the SBS-JC patients. Interestingly, malnutrition and sarcopenia were absent with the GLP-2-analogue TED, which should be considered in the postsurgical treatment, especially for those with ≤60 cm of SB. As reduced muscle mass was also common, the assessment of the body composition may be useful, including the muscle quality. While the preserved muscle strength is reassuring, the restoration of the muscle mass and quality should be targeted not only during, but also after weaning. Indeed, patients with malnutrition could benefit from more regular follow-ups. Longer-term studies are needed in order to assess the potential benefits of nutritional and physical interventions and associations with outcomes in chronic IF patients.

## Figures and Tables

**Figure 1 nutrients-15-00338-f001:**
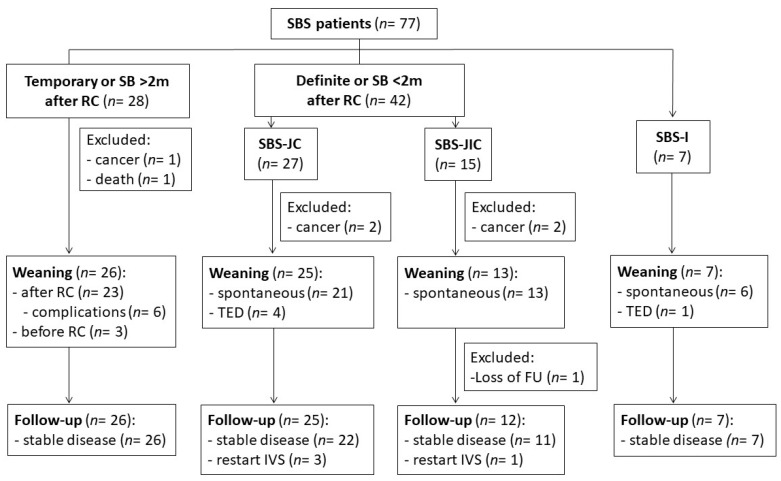
Flowchart of SBS patients during IVS and FU after weaning. Continuity surgery (RC) was performed in temporary and definite SBS patients with a colon-in-continuity. Patients weaned on TED (four SBS-JC, one SBS-I patients) were analyzed separately. Abbreviations: follow-up (FU), intravenous support (IVS), small bowel (SB), short bowel syndrome (SBS), ileostomy (SBS-I), jejuno-colic (SBS-JC) or jejuno-ileal anastomosis (SBS-JIC), restoration of continuity (RC), and teduglutide (TED).

**Figure 2 nutrients-15-00338-f002:**
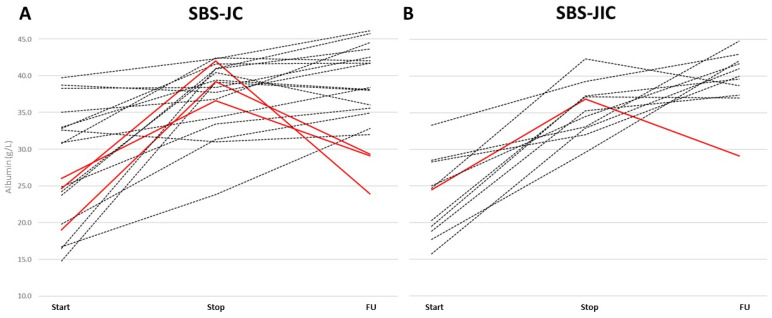
Individual changes in albumin for SBS-JC and SBS-JIC patients during IVS and FU. (**A**) SBS-JC and (**B**) SBS-JIC patients restarting IVS during FU are indicated in red (albumin <30 g/L during FU). Abbreviations: follow-up (FU), short bowel syndrome (SBS), jejuno-colic anastomosis (SBS-JC), and jejuno-ileal anastomosis (SBS-JIC).

**Figure 3 nutrients-15-00338-f003:**
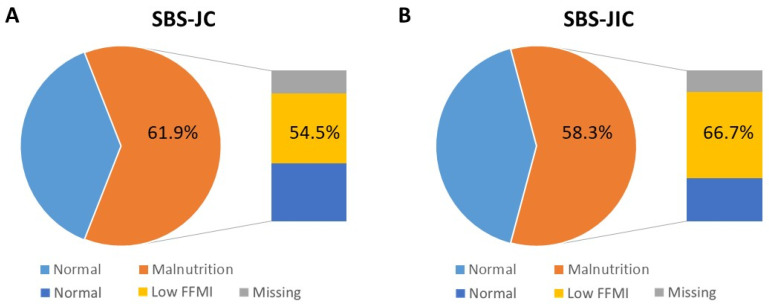
Prevalence of malnutrition in SBS-JC and SBS-JIC patients during FU after weaning. (**A**) SBS-JC and (**B**) SBS-JIC patients with malnutrition (GLIM-criteria), with prevalence of low muscle mass or FFMI in those with available BIA. Abbreviations: fat-free mass index (FFMI), short bowel syndrome (SBS), jejuno-colic anastomosis (SBS-JC), and jejuno-ileal anastomosis (SBS-JIC).

**Table 1 nutrients-15-00338-t001:** Demographics and clinical characteristics of IF patients at the start of IVS.

Variable		Total (*n* = 98)
Age (years)	Median (IQR)	54.8 (43.1; 65)
Gender	Female (%)	54 (55)
Weight (kg)	Median (IQR)	61 (52.2; 72)
BMI (kg/m^2^)	Median (IQR)	24.1 (21.7; 31.9)
Mechanism of IF	SBS (%)	77 (78.6)
	Mot (%)	9 (9.1)
	O (%)	5 (5.1)
	MD (%)	4 (4.1)
	F (%)	3 (3.1)
Severity of IF (max)	FE (%)	5 (5.1)
	PN1 (≤1 L)	9 (9.1)
	PN2 (1–2 L)	46 (46.9)
	PN3 (2–3 L)	16 (16.3)
	PN4 (>3 L)	22 (22.4)
IVS duration (weeks)	Median (IQR)	30.8 (13.7; 58.3)
FU duration (weeks)	Median (IQR)	40.1 (14.7; 69.9)

Mechanism and severity (daily mean of the maximum volume infused per week) of IF were defined according to ESPEN guidelines. Abbreviations: fistula (F), fluids and electrolytes (FE), follow-up (FU), intestinal failure (IF), intravenous support (IVS), mucosal disease (MD), motility disorder (Mot), obstruction (O), parenteral nutrition (PN), and short bowel syndrome (SBS).

**Table 2 nutrients-15-00338-t002:** Demographics and clinical characteristics of SBS patients.

Class	Variable	Temporary SBS (*n* = 26)	SBS-JC (*n* = 21)	SBS-JIC (*n* = 12)	SBS-I (*n* = 6)	SBS + TED (*n* = 5)	*p*-Value
Demo-graphics	Age (years)	58 (49; 69)	64 (45; 74)	57 (43; 69)	50 (32; 63)	56 (47;64)	0.57
	Gender (female)	13 (50)	17 (81)	8 (61.5)	1 (16.7)	1 (20)	0.02
Etiology (%)	Ischemia (A/V)	11/6 (65.4)	10/0 (47.6)	3/4 (53.8)	1 (16.7)	1/1 (40)	0.25
	IBD (CD/UC)	6/1 (26.9)	4/0 (19)	1 (7.7)	5 (83.3)	0 (0)	<0.01
	Radiation	1 (3.8)	3 (14.3)	2 (15.4)	0 (0)	0 (0)	0.48
	Surgical/adhesion	1 (3.8)	3 (14.3)	2 (15.4)	0 (0)	3 (60)	0.01
	Trauma/volvulus	0 (0)	1 (4.8)	1 (7)	0 (0)	0 (0)	0.65
IF class (%)	FE	2 (7.7)	0 (0)	0 (0)	3 (50)	0 (0)	<0.001
	PN1 (≤1 L)	2 (7.7)	2 (9.5)	0 (0)	0 (0)	0 (0)	0.69
	PN2 (1–2 L)	10 (38.5)	8 (38.1)	4 (30.8)	3 (50)	0 (0)	0.46
	PN3 (2–3 L)	8 (30.8)	4 (19)	2 (15.4)	0 (0)	1 (20)	0.50
	PN4 (>3 L)	4 (15.4)	7 (33.3)	7 (53.8)	0 (0)	4 (80)	<0.01
	Total SB (cm)	300 (240; 330)	105 (70; 160)	160 (70; 190)	250 (200; 300)	75 (69; 80)	<0.0001
Anatomy	Ileum (cm)	120 (13; 235)	0 (0; 0)	50 (25; 120)	NA	0 (0; 10)	<0.0001
	Colon (%)	100 (86; 100)	75 (70; 90)	100 (100; 100)	NA	80 (63; 88)	<0.0001
Duration	IVS (weeks)	10.5 (7; 14)	40 (28.5; 67)	23 (14.5; 64)	62.5 (35; 141)	166 (93.5; 376)	<0.0001
	FU (weeks)	29 (15; 48)	39 (12.5:84.5)	22 (15; 83.5)	40.5 (9; 45)	137 (76; 156)	0.04

Continuous data were reported as median (IQR) and categorical data as proportions (%) and analyzed using Kruskal–Wallis and chi-square tests, respectively. Abbreviations: arterial (A), Crohn’s disease (CD), fluids and electrolytes (FE), follow-up (FU), inflammatory bowel diseases (IBD), intestinal failure (IF), intravenous support (IVS), parenteral nutrition (PN), small bowel (SB), short bowel syndrome (SBS), ileostomy (SBS-I), jejuno-colic anastomosis (SBS-JC), jejuno-ileal anastomosis (SBS-JIC), teduglutide (TED), ulcerative colitis (UC), and venous (V).

**Table 3 nutrients-15-00338-t003:** Changes in nutritional markers during IVS and FU after weaning.

Variable	Time	Temporary SBS (*n* = 26)	SBS-JC (*n* = 21)	SBS-JIC (*n* = 12)	SBS-I (*n* = 6)	SBS + TED (*n* = 5)	*p*-Value
BW (kg)	Start IVS	71.4 (55.8; 81)	**65 (54; 74)**	58.6 (52.2; 83)	60.7 (56; 65)	55 (47; 67)	
	Stop IVS	**73.8 (60; 79.3)**	**68.8 (57; 79.8)**	65.5 (54; 77.4)	76 (64.3; 79.3)	66 (59.8; 71)	0.02
	FU	**77 (61.5; 87.3)**	**62 (53.5; 72)**	66 (54.5; 76.3)	73 (62.4; 80.5)	69 (57.5; 71.5)	0.0001
BMI (kg/m^2^)	Start IVS	24 (20.4; 28.8)	**23.6 (18.8; 28.8)**	21.1 (18.4; 30.5)	17.8 (17; 20.4)	18.7 (17.5; 20.9)	
	Stop IVS	**24.5 (22.2; 27.8)**	**24.7 (22; 29.6)**	23.2 (20.1; 27.4)	23 (20.8; 24.6)	22.3 (21.6; 22.8)	0.03
	FU	**26.9 (23.2; 31.2)**	**22.6 (20.9; 25.7)**	22.6 (20.3; 28.7)	22 (19.8; 24.4)	22.9 (20.7; 23.4)	0.0001
Albumin (g/L)	Start IVS	**23.4 (20.3; 28)**	**25.5 (20.7; 33)**	**23.7 (19; 27.5)**	**26.7 (22.4; 38)**	32.1 (28.3; 32.6)	
	Stop IVS	**36.3 (32; 40.9)**	**38.8 (34.9; 41)**	**34.9 (32.2; 37.3)**	**39.9 (37.2; 41.9)**	43 (36.4; 48.6)	0.91
	FU	**39.2 (36.5; 44.4)**	38.1 (32.8; 42.5)	40 (37.4; 42.1)	40.3 (38.8; 42.2)	40.9 (37.9; 45.8)	0.09
CRP (mg/L)	Start IVS	**8 (2.8; 17)**	**2 (1; 11)**	3.5 (2; 16.8)	8.5 (2.5; 24.8)	8 (1; 26)	
	Stop IVS	**2.5 (1; 8)**	**1 (1; 2.9)**	5 (2.3; 6.8)	1 (1; 13)	1 (1; 6.5)	0.47
	FU	1 (1; 6.8)	1 (1; 3)	1 (1; 7)	4 (2; 24)	1 (1; 1)	0.4

Significant within-group changes are marked in bold (see text for significance), *p*-value is given for between-group differences (ANOVA) in changes during IVS (start-stop IVS) and after weaning (stop IVS-FU). Abbreviations: body mass index (BMI), body weight (BW), C-reactive protein (CRP), follow-up (FU), intravenous support (IVS), short bowel syndrome (SBS), ileostomy (SBS-I), and jejuno-colic (SBS-JC) or jejuno-ileal anastomosis (SBS-JIC).

**Table 4 nutrients-15-00338-t004:** Micronutrients in SBS patients restarting IVS during FU.

Type	TS (%)	Ferritin (µg/L)	FA (nmol/L)	Vit B_12_ (pmol/L)	Vit A (µmol/L)	Vit D (pmol/L)	Vit E (µmol/L)	Cu (µmol/L)	Se (µmol/L)	Zn (µmol/L)	CRP (mg/L)
SBS-JC	**3**	**13**	19.1	334	**1.1**	**22.4**	25.9	**9.4**	**0.8**	25.5	1
SBS-JC	**13**	140	27.6	368	**1**	**28.6**	**21**	17.3	1.1	16.3	12
SBS-JC	50	126	11.5	230	**1.1**	**<10**	**18.4**	**6.3**	**0.9**	18.7	1
SBS-JIC	86	583	9.8	441	**0.7**	**61.8**	23.8	**7.5**	**0.6**	**8.4**	3
CIPO	20	235	27.4	**107**	2.78	**37.7**	32.8	**11.27**	1.29	12.57	1
Normal values	20–40	26–388	6.8–45.4	139–651	1.5–2.7	75–200	21–35	12.7–22.2	0.9–1.5	12.5–18	<6

Abnormal values are indicated in bold. Levels of 25-hydroxyvitamin D are given. Abbreviations: chronic intestinal pseudo-obstruction (CIPO), copper (Cu), folic acid (FA), short bowel syndrome (SBS), jejuno-colic (SBS-JC) or jejuno-ileal anastomosis (SBS-JIC), selenium (Se), transferrin saturation (TS), vitamin (vit), and zinc (Zn).

## Data Availability

Data are available upon reasonable request to the corresponding author.

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
