# Peer review of "Malnutrition with Low Muscle Mass Is Common after Weaning off Home Parenteral Nutrition for Chronic Intestinal Failure"

_nutrients, 2023, doi:10.3390/nu15020338_

Round 1

Reviewer 1 Report

Not sure that patients with narcotic bowel or small bowel diverticulosis really merit inclusion.  Very cloudy to the reader how you calculated your medians becuase not clear how many data-sets exactly each patient contributed to analysis due to the variable follow-ups done where some patient seen only once, others twice or 3 times.

Study Design Foundations:

Concern for data-heaping – precise and coarse data-points intermingled.

It is cloudy to the reader just how many sets of data points each patient would have… on Pg 3, lines 133-135 – It is stated that median follow-up was 41 weeks.  However, it is stated in line 101, patient were evaluated every 3-6 months (12 to 24 weeks).    What percentage of patients had only one set of data-points?   That means whatever empirical management advice or changes implemented for those patients would not be captured as they outstripped the second  data collection

Patient’s with only one set of data-points should be removed from the analysis.  Arguably patient with three sets of data-points should be analysed separately from two sets of data points.  The reason for this is that changes in management plan post-clinic visit are being differentially assessed – coming under scrutiny with different frequencies.  The return to clinic so soon after the 1st follow-up in some case may be directly linked to that 1st follow-up visit which means it is not necessarily a discrete occurrence in and it-self is a form of more precise intervention and precise monitoring.   By contrast, patients with only one-data set and “no –return” to clinic for the study duration have very coarse observations. The would appear to be a form of “data-heaping”.

Patient Population:  A subset of Crohns’ patients may get a salubrious effect from the Teduglutide which gives the some therapeutic remission of Crohns.   This could be a variable effect.  Could you do a sensitivity-analysis and pull-out those patient receiving maintenance-therapy for Crohns disease simultaneously with Teduglutide and being weaned?

The parameter Hand Grip Strength:   The sensitivity of this is up for debate as it is under conscious control of the patient.  It is unclear if there is any utility in the inclusion of this parameter.  Can you make a methodological argument to support?  Same thing for pre-albumen – see below.

The dysmotility patients need to carved out more in the text and have their own Table – why is there no Table with the demographics and clinical characteristics of the Dysmotility patients? 

Pg 2, Line 76.  Were there any empirical dose reductions for side-effects like diaphoresis or insomnia?

Page 6, Table 3.   The inclusion of Pre-albumen invokes very much consternation – especially in patients eating by mouth.  If for any reason a patient were to just say skip breakfast and then go to the lab, a vertiginous drop in the PreAlbumen lab value would be seen.  What is the reason that such a highly-sensitive parameter is included when the timeline for follow-up stretches out into months?

Page 7, line 225 ,,,  “stable CRP..”   need to see the raw data with the range… a CRP can be stable over time yet remain elevated….  An elevated CRP will govern how to interpret the trace elements – see below  ..  I am afraid this will have to mean a table in itself outlining the CRP across the whole board…

Pg 7, Table 4.

Please include a column with median CRP ( if available) in order to guide how tointerpret the Cu, Se and Zn in case as all of these are acute phase reactants and fluctuate. 

Page 8, Line248 – Did you spell-out in words GLIM previously ?

Pg. 10, Line 354 –In the dysmotility group of a whole, if malnutrition was common, what was the median frequency of follow-up  and is there a case to be made to better define the optimal frequency for follow-up?

Conclusions:  See above questions about the frequency of follow-up for malnourished patients who had to restart IVFS.  Can you generate any recommendations about what the frequency of follow-up should be for malnourished patients?

Reviewer 2 Report

This longitudinal study assesses the prevalence of malnutrition and sarcopenia among SBS-patients. It also aims to evaluate the nutritional and functional markers during and after weaning off HPN among chronic IF patients.

Ethics Committee Permission is missing.

The name of the y-axis is missing in Figure 2. In Table 4, vitamin B12 is not correctly marked. Data should be expressed as standard deviations, or standard errors not as medians.

The discussion section is unstructured, which makes it difficult to understand.

Author Response

This longitudinal study assesses the prevalence of malnutrition and sarcopenia among SBS-patients. It also aims to evaluate the nutritional and functional markers during and after weaning off HPN among chronic IF patients.

Ethics Committee Permission is missing.

We confirm that inclusion in the database was only possible after patients were individually informed and did not object that their data would be used for research. All data were collected during routine medical examination and patients did not undergo any modification of treatment or visits according to the protocol (non-interventional setting), with approval by the Institutional Review Board of HUPNVS, Paris 7 University, AP-HP France (see page 11, lines 403-404).

The name of the y-axis is missing in Figure 2.

We thank the reviewer for this comment and have added the y-axis (albumin (g/L)).

In Table 4, vitamin B12 is not correctly marked.

Thank you, we have changed this accordingly.

Data should be expressed as standard deviations, or standard errors not as medians.

We confirm that medians were used with non-parametric testing to account for not normally distributed variables (see page 3, lines 117-8). This was especially relevant for smaller numbers of subjects in some analyses, and possibility of outliers (eg. variables as CRP).

The discussion section is unstructured, which makes it difficult to understand.

We agree with the reviewer that a more structured outline is helpful and have done this by adding subtitles, in order to guide the reader through the discussion.

Round 2

Reviewer 1 Report

How do you explain the disparity between increased Hand Grip Strength in general  yet  patients experienced decreased PA during F/U in those restarting IVS?  This raises questions about the robustness of Hand Grip Strength...and certainly the messaging may be to centers that don't have HGS, to not worry about acquiring it... there needs to be some amplification in the discussion about the limitations of the applicaton of HGS

Author Response

Thank you for this comment, we clarified that even if HGS was normal, the increase during FU was only found in those with no need to restart IVS (so after exclusion of those restarting IVS, see page 8, lines 267-269). We agree with the reviewer that even if increased handgrip strength during FU may be of interest (and motivating for patients), it is not of key importance in the routine evaluation. Thus, we have emphasized it’s limited value and the fact that it is only supportive and not core phenotypic criterion of malnutrition in the discussion on page 8, lines 384-386). Future studies should indeed focus on body composition and muscle quality or phase angle.